# Grade Progression and Intrapatient Tumor Heterogeneity as Potential Contributors to Resistance in Gastroenteropancreatic Neuroendocrine Tumors

**DOI:** 10.3390/cancers15143712

**Published:** 2023-07-21

**Authors:** Diana Grace Varghese, Jaydira Del Rivero, Emily Bergsland

**Affiliations:** 1Developmental Therapeutics Branch, National Cancer Institute, National Institutes of Health, Bethesda, MD 94158, USA; diana.varghese@nih.gov (D.G.V.); jaydira.delrivero@nih.gov (J.D.R.); 2UCSF Helen Diller Family Comprehensive Cancer Center and Department of Medicine, University of California, San Francisco (UCSF), San Francisco, CA 94158, USA

**Keywords:** gastroenteropancreatic (GEP) neuroendocrine neoplasm (NEN), grade, neuroendocrine tumor (NET), differentiation, heterogeneity, grade progression, grade migration, neuroendocrine carcinoma, Ki-67 index

## Abstract

**Simple Summary:**

Neuroendocrine neoplasms (NENs) have a varied biology. While well-differentiated neuroendocrine tumors (NETs) typically demonstrate indolent behavior, a subset of these tumors exhibits rapid growth, and eventual resistance to various treatments over time is the rule. We discuss intratumoral heterogeneity as a potential contributor to tumor progression and treatment resistance in these tumors. Clinicians should be aware of the potential for intratumoral heterogeneity, including variable somatostatin receptor (SSTR) expression, molecular alterations, Ki67 indices, and grade progression over time. Histopathologic heterogeneity can exist at baseline within primary tumors, metastases, or between a primary tumor and synchronous metastases. In addition, some NETs demonstrate increases in Ki67 index and/or grade over the disease course. Apparent reductions in tumor grade over time should be interpreted with caution, considering the potential impact of prior treatment and the biopsy size and site. Future strategies may include incorporation of liquid biopsies, serial tumor biopsies, and imaging techniques to identify the most highly proliferative regions for biopsies (recognizing that a Ki-67 index obtained from a routine FNA or core needle biopsy may not adequately represent the whole tumor). Long term, a better understanding of the mechanisms underlying tumor heterogeneity and resistance to therapy will help guide a personalized approach to patients with GEP-NETs.

**Abstract:**

Gastroenteropancreatic neuroendocrine neoplasms (NENs) are a heterogenous group of tumors that are incurable when metastatic, regardless of grade. The aim of this article is to understand tumor heterogeneity and grade progression as possible contributors to drug resistance in gastroentropancreatic neuroendocrine tumors (GEP-NETs). Heterogeneity has been observed in the genetic, pathological, and imaging features of these tumors at baseline. Diagnostic challenges related to tumor sampling and the potential for changes in grade over time further confound our ability to optimize therapy for patients. A better understanding of NEN biology and tumor heterogeneity at baseline and over time could lead to the development of new therapeutic avenues.

## 1. Introduction

Neuroendocrine neoplasms (NENs) are a diverse group of malignancies that originate from the neuroendocrine enterochromaffin cells distributed throughout the body. The most common sites are the gastrointestinal tract, pancreas, and lungs. Their clinical course varies from slowly progressing over many years to having a highly aggressive disease biology. The nomenclature of NENs has evolved in the past few decades to incorporate the difference in tumor biology and behavior. In 2017 (pancreas) and 2019 (gastrointestinal tract), the World Health Organization (WHO) updated the classification schemes for gastroenteropancreatic neuroendocrine neoplasms (GEP-NENs) to address the fact that high-grade (G3) disease includes both well and poorly differentiated tumors [1,2]. As such, NENs are now classified as well-differentiated neuroendocrine tumors (NETs)—Grade (G)1, G2, and G3 NET or G3 poorly differentiated neuroendocrine carcinoma (NEC), based on their histopathologic features, differentiation, mitotic count, and Ki-67 proliferation index (Table 1). Histopathologic classification is not always straightforward; thus, the 2022 WHO classification acknowledges the growing importance of immunohistochemical staining for somatostatin receptors (SSTRs) 2/5, Rb, p53, ATRX, and/or DAXX to distinguish between G3 NETs and NEC. DLL3 and ODF1 expression are other emerging biomarkers that may have value in distinguishing between G3 NETs and NEC [3].

The Ki-67 index is a marker of cellular proliferation. G1 NETs require a Ki-67 index of <3% and a mitotic rate of <2 mitosis/2 mm^2^, G2 is defined by a Ki-67 index of 3–20% or a mitotic rate of 2–20/2 mm^2^, and G3 NETs and NEC have a Ki-67 index of >20% or a mitotic rate of >20 mitosis/2 mm^2^. Well-differentiated G3 NETs and poorly differentiated NEC are distinguished based on their morphological and immunohistochemical tumor characteristics. 

The five-year survival varies significantly in terms of the Ki-67 index, even within grades: 81% [for a Ki-67 index of 3–5%], 72% [for a Ki-67 index of 6–10%], 52% [for a Ki-67 index of 11–20%], 35% [for a Ki-67 index of 21–50%], and 22% [for a Ki-67 index of 51–100%], respectively [5]. Thus, the Ki-67 index should be considered to be a continuum, and further subdivision might be necessary to further refine the existing WHO classification [5]. Even in poorly differentiated NECs, the Ki-67 index may matter; tumors with a Ki-67 index of <55% have a better prognosis than those with a Ki-67 index of >55% [6].

Recently, data have suggested that some GEP-NENs demonstrate intrapatient heterogeneity at baseline and over time. Heterogeneity exists at many levels, including differences in the Ki-67 index and even the grade within a primary tumor, between a primary tumor and a metastasis, between synchronous metastases, and in metachronous metastases over time [7]. In many cases, grade progression occurs, meaning a higher grade is identified in subsequent biopsy material [8]. In others, an increase in the Ki67 index occurs without an overt change in grade (i.e., grade migration). Heterogeneity at the molecular level and in the context of functional imaging has been noted [9,10,11]. Taken together, the data suggest that intratumoral heterogeneity can exist within a patient at a given timepoint and emerge over time, potentially contributing to resistance and confounding our ability to optimally risk stratify patients and individualize therapy. The precise role of intrapatient tumor heterogeneity in mediating treatment resistance has not been elucidated but remains an area of active study.

## 2. Clinical Features and Treatment of Low-Grade Neuroendocrine Tumors

Low-grade NETs encompass well-differentiated G1 and G2 tumors. Recurrent genomic alterations are uncommon in gastrointestinal NETs, but pancreatic NETs are characterized by alterations in *MEN1*, *DAXX/ATRX,* and mTOR pathway genes [10,12]. Low-grade GEP-NETs are relatively indolent with a median survival measured in years, even when metastatic. Boyar et al. reported a five-year survival of 53.9% in well-differentiated pancreatic NETs (70.2% for local and 33% for distant disease at diagnosis), and Dasari et al. reported a five-year survival of 50% for distant G1/G2 pancreatic NETs and 69% for distant G1/G2 small intestine NETs [13,14]. The primary treatment modality for local disease is complete surgical resection if feasible. Surgical cytoreduction is also performed in selective patients with metastases. While data from randomized trials are lacking, the resection of more than 70–90% of liver metastases is associated with improved overall survival (OS) and progression-free survival (PFS) in patients with GEP-NETs [15,16].

Somatostatin analogs (SSAs), such as octreotide and lanreotide, are used for symptom control from excess hormone production [17] and as an anti-proliferative therapy in advanced tumors [18]. Data from a CLARINET study demonstrated that lanreotide delays progression in nonfunctioning GEP-NETs with a Ki-67 index up to 10% [19]. The cytostatic benefit of long-acting octreotide in midgut NETs was demonstrated in a PROMID study [20].

In the face of subsequent disease progression, additional therapies can be considered for GEP-NETs, including treatment with everolimus [21,22], a mammalian target of rapamycin (mTOR) inhibitors, and peptide receptor radionuclide therapy (PRRT) with ^177^Lu-DOTA^0^-Tyr^3^-octreotate (^177^Lu-DOTATATE) [23,24]. Liver-directed therapies, such as liver ablation and intra-arterial embolization, are routinely used for liver-dominant disease, particularly when it is necessary to alleviate symptoms related to pain, tumor bulk, and/or excess hormones [25]. Chemotherapy (e.g., temozolomide plus capecitabine) has a limited role in the treatment of gastrointestinal NETs, but is routinely used in pancreatic NETs given a response rate of 30–40% [26,27]. Furthermore, treatment with sunitinib, a vascular endothelial growth factor (VEGF) receptor tyrosine kinase (RTK) inhibitor, is approved for pancreatic but not GI NETs [28].

Importantly, the optimal sequence of therapy in GEP-NETs has not been defined. Treatment is individualized based on many factors, including the tumor’s location, stage, extent, rate of growth, Ki-67 proliferation status, somatostatin receptor expression, functional status, patient preference, and comorbidities [29,30]. Multidisciplinary decision making facilitates the integration of inputs from surgical oncology, medical oncology, nuclear medicine, and interventional radiology.

## 3. Clinical Features and Treatment for High-Grade Neuroendocrine Neoplasms

### 3.1. Grade 3 Neuroendocrine Tumors

G3 NETs account for 5.6–9% of all GEP-NENs, 12% of well-differentiated tumors, and roughly 12% of all high-grade NENs, with the pancreas being the most common primary tumor site [31,32,33,34,35]. About 82–92% of G3 NETs demonstrate somatostatin positivity in functional imaging [33,36,37]. G3 NETs are associated with a better median OS than that of poorly differentiated NEC (33–99 months [31,36] vs. <12–17 months) [14,35]. The genomic features of G3 NETs are not well defined; however, emerging data suggest that pancreatic tumors may be enriched for alterations in *TP53*, *SETD2*, and *BRAF* [10]. Variable rates of alteration in *MYC* and *Rb1* have been reported [10,38,39].

Treatment guidelines for G3 NETs are evolving. The NCCN and ESMO guidelines recommend treatment that mirrors that of low-grade NETs or NEC depending on clinical and molecular features [23,40]. The resection of the primary tumor and a regional lymphadenectomy are recommended for localized disease with a favorable biology. In contrast, if the tumor has a Ki-67 index > 55% and/or is negative in DOTATATE imaging (has an unfavorable tumor biology), platinum-based chemotherapy may be appropriate up-front, even if the tumor is localized. For advanced disease, current guidelines [23,40] recommend chemotherapy (including capecitabine/temozolomide, oxaliplatin-based chemotherapy, or platinum/etoposide), PRRT, sunitinib (pancreas only), everolimus, or liver-directed therapy depending on the disease extent, Ki-67 index, and tumor biology. Somatostatin analogs have not been well studied in G3 NETs for tumor control, but patients can likely benefit from their anti-secretory properties if a tumor is functional [37,41]. The results of multiple retrospective studies have shown that platinum-based chemotherapy has a lower objective response rate in well-differentiated G3 NETs compared to that of NEC, particularly when the Ki-67 index is <55% [42,43]. The use of combination immunotherapies has been shown to have an objective response rate of around 15% in high-grade NENs, although the precise level of activity in G3 NETs is not well-defined [44,45].

### 3.2. Grade 3 Neuroendocrine Carcinomas

NECs are characterized by either small-cell or large-cell morphology, a poor prognosis, and rapidly progressive disease [3,46]. NECs most commonly arise in the lungs, where the disease is smoking-related and a small-cell histology dominates. Extrapulmonary NECs represent approximately 10% of all NEC cases overall, with the majority originating in the gastrointestinal system (37.4%) and unknown sites (28.2%) [47]. The small-cell or large-cell subtypes predominate depending on the organ site. Most patients with extrapulmonary NECs present with metastatic disease, including more than two-thirds of those arising in gastrointestinal sites [47]. NECs are characterized by genomic instability and enriched for alterations in *TP53* and *Rb1* [39], along with mutations often seen in non-neuroendocrine tumors of the same organ site, including *KRAS*, *BRAF*, *PTEN*, *PI3KCA*, and *APC* [48]. The Ki-67 index of these tumors is usually >70%.

The median survival of NECs varies by organ site but is generally less than 12 months; for GEP-NECs, the median OS is 7.5 months (25.1 months for small intestine vs. 5.7 months for pancreatic primaries) [47]. Extrapolating from small-cell lung cancer, platinum/etoposide (with a response rate of 30–50%) is generally employed as a first-line therapy [49]. Salvage therapy is inadequate, with poor outcomes regardless of choice of chemotherapy, although irinotecan-based therapy is emerging as a potential option for GEPNEC [50,51]. A retrospective analysis of patients who received various second-line chemotherapy regimens showed no difference in OS regardless of the drug used [52]. Limited data suggest that combination immunotherapy with ipilimumab and nivolumab has activity in a subgroup of patients with G3 GEP-NENs, with some studies pooling G3 NETs/NECs [44,45,53]. Combination immunotherapy with durvalumab and tremelimumab showed limited activity overall in a multi-cohort DUNE study, although the primary endpoint of the patients’ nine-month OS was achieved in the G3 GEP-NEN cohort despite a low response rate [54]. Further investigations with CAR-T cells and bispecific antibodies are underway [55].

## 4. Diagnostic Challenges Related to GEP-NEN Classification

### 4.1. FNA vs. Core Biopsy vs. Surgical Sample

The Ki-67 index appears to be more likely to identify a grade change than the mitotic index, recognizing that the mitotic rate requires a larger sample and the Ki-67 index is easier to assess in smaller samples [8]. In surgically resected liver metastases, tumor heterogeneity is more common in higher-grade tumors (i.e., G2 and higher). Measuring the highest grade (“hot spot”) in a core biopsy sample most accurately represents the tumor grade of the whole sample [8].

Endoscopic ultrasound (EUS)-guided techniques are evolving, with fine-needle aspiration (FNA) (with a sensitivity of 80–90% and a specificity of about 95%) and larger-bore fine-needle biopsies (FNB) increasingly being used to diagnose pancreatic NETs [56,57,58,59]. In one study of N = 51 patients with panNETs, the sensitivity of EUS-FNA was 89.2% [60]. Grade concordance between EUS-FNA and the surgical specimens was observed in 69.2% (9/13) of the patients. Interestingly, the concordance rate was higher in tumors <20 mm (87.5%; 5/6) compared to that of tumors ≥20 mm (57.1%; 4/7), likely reflecting the tumor heterogeneity not appreciated by FNA. Higher-grade (G2/G3) tumors were more likely to be large (>20 mm), have main pancreatic duct (MPD) obstructions, and appear heterogeneous in EUS than G1 tumors [60]. In another study comparing EUS-FNA with surgery, grade classification discordance was especially common in G2 tumors, with cytology underestimating the grade at surgery in 10/14 tumors (71%) [61]. Three additional G1/2 tumors were upgraded to G3 at surgery, while 2/7 G2 tumors were downgraded to G1 at resection [61]. EUS-guided larger-bore biopsies (with a 19-gauge needle) represent a potential advance, with samples adequate for histologic diagnosis in 93% of cases and grade classification concordance with surgery in 83% of (N = 12) paired panNET samples. Only one patient demonstrated upgrading (G1 to 2) or downgrading (G2 to G1) at surgery [58]. Taken together, the data suggest that EUS-guided biopsy techniques are evolving but are generally effective for the diagnosis of panNETs. However, accurate grade classification can be challenging, particularly for large or G2 tumors, or when the sample is small.

### 4.2. G3NENs with Ambiguous Morphology

The revised WHO classification of NENs requires morphological diagnoses to differentiate between G3-NETs and NEC. However, classification based on morphology has proven challenging, with significant discordance between reviewers noted in some, but not all studies [34,35]. The Ki-67 index has the appeal of being an objective measure; however, numerous studies suggest that the Ki-67 index is affected by both intrapatient and interobserver variability [62]. Furthermore, there is significant overlap in the Ki-67 index observed in G3 NETs and NEC, such that the proliferation index alone is not diagnostic [35,43]. Tang et al. in 2016 examined 33 cases of G3 pancreatic NENs and identified both well-differentiated NET (G3 NET) and poorly differentiated NEC subtypes [63]. The morphologic and immunohistochemical (IHC) findings of these tumors (either from a surgical resection or core biopsy) were reviewed by three different pathologists, and the objective of the study was to assess the concordance of the morphologic findings. Discordance amongst the pathologists was noted in 61% of the cases; in the remaining cases, there was a consensus as to differentiation (well-differentiated NET or poorly differentiated NEC). Subsequent studies have reported variable discordance between pathologists when classifying G3 NENs, although it seems to be most common when discriminating between well-differentiated G3 NETs and large-cell NEC [35,64]. The distinction is important, recognizing that G3 NETs are associated with improved survival compared to their NEC counterparts [35,63,64].

Given the challenges associated with the classification of high-grade NENs (G3 NETs/NEC), the integration of additional molecular biomarkers, such as Rb, p53 ATRX, DAXX, and menin, has been recently proposed as a means of generating a more accurate diagnosis [3]. Tang et al. suggested that the incorporation of additional clinical information and biomarkers (such as IHC for ATRX, DAXX, p53, and Rb) could lead to a more accurate diagnosis compared to morphology alone. Specifically, a prior history of low-grade NET, aberrant ATRX, or DAXX staining suggests a G3 NET; aberrant expressions of p53, a global loss of Rb or SMAD4, or coexisting adenocarcinoma are supportive of a NEC diagnosis [63]. Moreover, an organoid growth pattern, the absence of desmoplastic stroma, and a capillary network in direct contact to tumor cells are consistent with the diagnosis of a G3 NET [35]. The recent 2022 WHO classification of neuroendocrine neoplasms also supports the used of IHC markers to distinguish G3 NETs from NEC, with G3 NETs being more likely than NEC to share genomic and IHC features with G1/2 NETs [3]. While not definitive, a loss of menin, p27, ATRX or DAXX staining, and/or retained SSTR 2/5 expression support a G3 NET diagnosis. In contrast, a global loss of Rb, diffuse positivity or global loss of p53, and/or the loss of SSTR 2/5 suggest NEC [3]. The data suggest that NEC is a distinct entity and does not typically arise via progression from well-differentiated NETs [65]. In contrast, G3 NETs regularly appear to arise in the setting of a prior history of G1/2 NETs [3,66]. Distinguishing between the two subtypes of G3 GEP-NENs is important given clinically relevant differences in their outcome and treatment depending on the histologic type (G3 NET v NEC) [35,63].

## 5. Intrapatient Heterogeneity in GEP-NENs—Histopathologic Classification and Proliferation Rate

Heterogeneity in GEP-NENs has been described in the literature at various levels. These include differences in location, histopathological grade, hormone secretion, and somatic molecular and germline genetic changes [67]. As noted above, the Ki-67 proliferation index and tumor differentiation (well differentiated vs. poorly differentiated) have been shown to predict disease outcome and influence the choice of therapy. However, heterogeneity in the Ki-67 index and/or grade also exist within some individual patients at baseline, as well as over time, presenting a diagnostic challenge and suggesting a potential mechanism for variable tumor biology and resistance to treatment. It is not clear if the grade heterogeneity in individual patients is due to inherent phenotypic or genetic heterogeneity [68] or treatment effects [69].

### 5.1. Heterogeneity of Grade within Primary Tumors at Baseline

Heterogeneity in the Ki-67 index and/or grades within and between synchronous primary GEP-NETs is relatively common (Table 2). Grillo et al. examined 60 patients with a total of 250 GEP-NET tumor samples (93% G1/2); approximately 5% of the tumors demonstrated enough intratumoral variability in the Ki-67 index to change the grade of the primary tumor [7]. Eight patients had multiple primary tumors, and grade discordance between tumors was noted in 3/8 (37.5%) of the patients. The Ki-67 index correlated with tumor size, with 1–2 cm tumors showing a higher grade (G2) compared to subcentimeter lesions (G1) [7]. Grade discrepancy was observed in 18% of N = 28 patients with multi-focal small bowel NETs in a pathological analysis [70]. Similarly, Keck, et al. reported grade discordance between primary tumors in 24% of patients with synchronous (largely small bowel) tumors [71]. The areas of highest proliferative activity (“hotspots”) are routinely identified manually and used to assign grade in GEP-NENs, although digital image analysis has been proposed as an emerging technology for assessing intratumor heterogeneity [72].

### 5.2. Heterogeneity between Primary Tumors and Synchronous Metastases

Grade discordance between primary tumors and synchronous metastases has also been observed [7,68] (Table 2). The Ki-67 proliferative index frequently differs in primary and synchronous metastatic sites, and the variability is significant enough to cause a change in WHO grade in at least 23–34% of patients [7,71]. For example, Grillo et al. reported grade discordance (primary vs. metastasis) in 23% of cases (with a higher grade identified in synchronous metastases) [7]. Out of 11 cases with grade discordance as per the Ki-67 index, a discordant grade was also indicated by the mitotic index in six cases [7]. Keck et al. reported grade discordance in a third of resected primary and metastatic GEP-NETs [71]. In most cases of discordance, the grade was higher in the metastatic site (24%; usually G2 v G1); 10% demonstrated a lower grade in the metastatic site compared to the primary tumor [71]. Importantly, a higher grade in metastatic sites relative to primary sites is associated with worse clinical outcomes (e.g., the overall and progression-free survival rates) [71,73,74]. The expression of chromogranin A and synaptophysin also sometimes varies between primary and metastatic sites, highlighting another layer of heterogeneity in GEP-NETs [73].

### 5.3. Heterogeneity of Ki-67 in Synchronous Metastatic Deposits

The heterogeneity of the Ki-67 proliferation index in metastatic sites also frequently occurs, with grade discordance occurring both within a single metastasis and between synchronous metastases (Table 2). Yang et al. assessed the Ki-67 index in 45 surgically resected metastatic well-differentiated G1/G2 GEP-NET liver metastases [8]. Forty-seven percent of the liver metastases showed intratumoral heterogeneity. When the highest Ki-67 index (“hot spot”) was used instead of the mean Ki-67 index, the grade changed from G1 to G2 in 33.3% of the hepatic metastases (91% of the G2 tumors were found to show grade heterogeneity) [8]. When the synchronous resected GEP-NET metastases were compared, grade discordance was noted in 40% of cases [71]. Shi et al. reported that five of the eight cases of small bowel NETs with at least one G3 liver metastasis had evidence of all three grades simultaneously (in patients with multiple resected liver lesions), and the Ki-67 index and intratumoral heterogeneity increased with tumor size [74]. From a practical standpoint, the data support measuring the Ki-67 index in metastatic sites (not just primary tumors) and using “hotspots” to identify the regions with the highest Ki-67 index for grading. Importantly, a change in grade relative to primary tumors appears to be more likely in distant metastases as opposed to locoregional (nodal or mesenteric) metastases, with the highest risk in distant sites other than liver metastases [7].

**Table 2 cancers-15-03712-t002:** Selected studies assessing heterogeneity in synchronous biopsies in GEP-NETs: Ki67 proliferation index and/or grade.

Reference	N	Site of Origin	Time Between Biopsies (mo)	Type of Heterogeneity		Description	Grade Change, (n)
				**Ki67** **index** **N (%)**	**Grade** **N (%)**		
Grillo [7]	60	GEP-NET ^#^ (52% SB; 27% P)	Not defined	22/60 (37%)	3/60 (5%)	Discordance within primary tumor	G2 → G1 (2) (NOS) G3 → G2 (1) (NOS)
Grillo [7]	8		Not defined		3/8 (37%)	Discordance within multiple primary tumors	Larger tumors (1–2 cm) G2, smaller tumors G1
Grillo [7]	47		≤4 months	24/47 (51%)	11/47 (23%)	Variable Ki67 index +/− **higher grade** in synchronous metastases vs. primary tumors	G1→ G2 (8) (NOS) G2→ G3 (2) (both P)
Grillo [7]	60		Not defined		31/60 (52%)-distant 10/44 (23%)-locoregional	Grade discordance in metastases vs. primary tumors	
Keck [71]	103	Resected GEP-NET ^^^ (77% SB; 20% P)	Not defined-resected primary tumors with “concurrent” metastases		25/103 (24%) ^&^	**Higher grade** in liver or lymph node metastases vs. resected primary tumors	G1→ G2 (24) (NOS) G2→ G3 (1) (NOS)
					10/103 (10%) ^&^	**Lower grade** in metastases vs. resected primary tumors	G2 →G1 (10) (NOS)
					9/38 (24%) ^&^	Discordance in synchronous primary tumors (NOS)	(NOS)
					8/20 (40%)	Discordance in synchronous liver metastases	(NOS)
Yang [8]	41 (45 tumors)	Resected NET mix * (29% P; 27% SB)	Same day-liver metastases		21/45 tumors (47%)	Discordance (G1 and G2) within single liver metastases	91% of G2 cases heterogeneous (NOS)
Shi [74]	27 (188 liver lesions) 20 primary tumors	Resected Liver mets (SB)	≈50% synchronous		13/20 (65%)	Ki67 discordance in resected liver metastases ≥1 cm; Grade discordance between primary and liver metastases	N = 27 with liver mets, 10 (37%) only G1, 9 (33%) G2 +/-G1 tumors, 8 (30%) G3 +/- G1/2 G1→ G2 +/-G1 (6/17) G1→ G3 +/-G1/2 (5/17) G2→G2 (1/3) G2→ G3 (2/3)
Shi [73]	35	GEP NEN NET mix (27.7% P; 26.7% R)	71% synchronous	19/35 (54.3%)	4/35 (11%)	**Higher grade** in metastases vs. primary tumors	G1→ G2 (3/35) G1→G3 (1/35)
					1/35 (2%)	**Lower grade** in metastases compared to primary tumors	G2→G1 (1/35)

^#^ pancreas, jejunum/ileum >> stomach, duodenum, colon/rectum; grade at dx-N = 37 G1, N = 19 G2, N = 4 G3; ^^^ small bowel> pancreas >> duodenum, colon; * pancreas, small bowel >> rectum, lung, bile duct, stomach + unknown primary; N = number of patients (unless otherwise specified); GEP-NET = gastroenteropancreatic neuroendocrine tumor; SB = small bowel; P = pancreas; NET = neuroendocrine tumor; NOS= not otherwise specified; R = Rectum; ^&^ modified WHO.

### 5.4. Grade Progression and Grade Migration over Time

In addition to the existence of heterogeneity in the Ki-67 index within primary and metastatic tumors at a given time point, increases in the Ki-67 index (+/− grade) with the evolution of tumors over time have been reported (Table 3). Botling et al. studied 46 patients with pancreatic NET tumor samples collected more than a year apart [75]. The median Ki-67 index at diagnosis (from a mix of primary and metastatic lesions) was 7% (1–38%); all follow-up samples were from metastatic sites. In 34 paired samples, 76% demonstrated an increase in the Ki-67 proliferation index, with the median change in the Ki-67 index over time being +14% (with a range from −11 to +80%). A longitudinal grade increase (mostly G1/G2 to G3) was identified in 57.6%; a decrease was observed in 6.8%. Of note, most G3 tumors retained a well-differentiated morphology. Grillo et al. evaluated metachronous metastases in 12 patients with GEP-NETs. Grade concordance between the primary tumor and metastases was noted in 17%; a total of 83% showed enough of an increase in the Ki-67 index to change grades (nine out of ten switched from G1 to G2, and one demonstrated a conversion to a G3 NET compared to the primary tumor). Grade progression was more common in patients with metachronous metastases than those with synchronous metastases [7]. In another study of N = 43 patients with well-differentiated NETs of different primary sites (including lung) over time (with a time interval of 4–81 months), grade discordance was demonstrated in 37% [69]. The majority (75%) of these cases demonstrated grade progression (31% G1 to G2, 13% G2 to G3, and 31% G1 to G3); all G3 tumors retained a well-differentiated morphology. Of note, three cases of upstaging and three cases of downstaging occurred in the setting of the acquisition of a larger pathology sample (FNA to core, or core to surgery). Panzuto et al. reported on N = 43 patients with G1/G2 GEP-NETs and serial biopsies over time [76]. The metachronous biopsies revealed an increase in the Ki-67 index over time in 65% of the patients, translating into a higher grade in 28% of the cases (mostly panNETs). A grade decrease over time was noted in only two cases (5%), both of which arose in the small bowel. In a recent work by Merola et al. analyzed the Ki-67 index at baseline and after recurrence in a group of small bowl and pancreatic NEN patients; 34% of recurrences showed an increased grade, demonstrating how the Ki-67 index can change over the course of the disease [77].

With respect to G3 NETs specifically, studies have suggested that G3 NETs can occur de novo or emerge over time [31,36]. When it occurs over time, it is most commonly documented in patients with pancreatic NETs that were originally grade 1 or 2. The magnitude of the change in the Ki-67 index is more prominent in metachronous (especially non-hepatic) metastases and is associated with poor outcomes [7]. Prospective studies are needed to fully assess the incidence, but retrospective data suggest grade progression may contribute to resistance over time in patients with GEP-NETs.

### 5.5. Treatment-Related Changes in High-Grade Tumors

Grade migration in G3 neuroendocrine neoplasms has not been well studied. Serial Ki-67 index values were evaluated in one series of 20 treated G3 NENs (mostly pure or mixed NEC) [78]. In 90% of the cases, the Ki-67 index was lower in the post-treatment sample compared to that of the pretreatment sample (*p* < 0.02) [78]. In 15/20 of the samples, a Ki-67 index of <20% was observed at least focally, and in 6/20 of the cases, the final Ki-67 index assessment based on “hot spots” falsely suggested a low-grade (G1 or G2) tumor. In 45% of the cases, the Ki-67 index was heterogeneous, with “cold-spot regions” in the G1 or G2 range. In approximately 50% of the cases, the post treatment morphology was similar to that of the pre-treatment sample; four cases showed more nuclear pleomorphisms and seven showed evidence of a more well-differentiated morphology post therapy [78]. Taken together, the data suggest that intrapatient heterogeneity in the Ki-67 index is common post chemotherapy and/or radiation in NEC; thus, the post treatment classification and grading of treated G3 NENs can be unreliable. The clinical implications of a reduction in the Ki-67 index or morphologic changes post treatment are unclear. As such, the current consensus is that a pretreatment diagnosis of NEC or a G3 NET should be consistent throughout the disease course [78].

## 6. Heterogeneity in GEP-NENS: Molecular Features

### 6.1. Heterogeneity of Molecular Features—Primary vs. Metastasis

Walter et al. explored the genetic heterogeneity in small intestine G1/G2 NETs with tumor samples obtained from primary (small intestine) and synchronous liver metastases in five patients [79]. The somatic mutation rate was less than <1/Mb. Interestingly, however, the authors documented varying degrees of discordance in the copy number variants and single-nucleotide variants in the primary and metastatic samples, with one pair of tumor specimens having no common variants at all. The analysis of allele frequency in the metastatic site suggested that clonal selection from the primary site contributes to metastatic spread [79]. The whole-genome DNA methylation analysis also suggested that epigenetic alterations play a role in the tumorigenesis of small intestine neuroendocrine neoplasms [79], and this is consistent with a lack of recurring mutations.

Zhou et al. used single-cell RNA sequencing (scRNA-seq) to study the spatiotemporal heterogeneity in one patient with a G2 pancreatic NET [9]. Heterogeneity in the transcriptome between cells from the primary site, metastatic site, and within the tumor microenvironment was identified. Based on copy number variations (CNV), a few tumor-associated clusters were identified, representing genes encoding cell functions such as cell cycle proliferation and differentiation. Hypoxia pathway genes were expressed predominantly in the primary tumor, whereas gene clusters with metastatic potential and drug resistance were enriched in the metastatic sites. The expression of *PSCK1/SMOC1* was shown to correlate with a high risk of recurrence and was validated in a cohort of 30 patients with G1/G2 NETs [9].

### 6.2. Heterogeneity of Molecular Features—Changes over Time

Next-generation sequencing (NGS) has proven to be a valuable tool for studying the molecular underpinnings of NENs. Emerging data suggest that genomic alterations in NENs can evolve over time. Raj et al. analyzed tumor samples from 96 patients with metastatic panNETs. The tumor mutation burden increased with grade [10]. In twelve patients, the samples were analyzed longitudinally at multiple time points. Eight patients (67%) demonstrated grade progression over time, and some chemotherapy-treated patients demonstrated acquired tumor hypermutation. Moreover, tumor samples from three patients treated with everolimus showed evidence of clonal selection in the context of acquired resistance to therapy. The presence of an *SETD2* or *BRAF* mutation was associated with an aggressive phenotype [10]. Tang at al. conducted whole-exome sequencing of tumors from a single patient with a metastatic G2 pancreatic NET [66]. Tissue samples from the patient’s pancreas, liver, and lymph nodes were collected at various time points during disease progression over a course of thirteen years. Driver genes associated with tumor progression were identified in all tumor samples. However, the tumor mutation burden (TMB) in the synchronous and metachronous metastatic samples (liver and lymph nodes) was found to be lower than that of the primary tumor (pancreas). The analysis of the clonal and sub-clonal architecture and gene profiling indicated that most of the metastatic subclones could be tracked back to the primary tumor. The authors proposed two potential mechanisms for this metastatic spread: (1) the presence of a subclone from the beginning of the tumor origin; and (2) polyclonal seeding events followed by metastasis spread [66]. Additional studies are needed to fully understand the degree to which the molecular underpinnings of GEP-NENs change over time.

## 7. Heterogeneity in GEP-NENs—Functional Imaging with ^18^F-FDG (18-Fluorine Fluorodeoxyglucose) vs. ^68^Ga DOTATATE PET

Somatostatin receptor (SSTR) positron emission tomography (PET) imaging with ^68^Ga-DOTATATE and ^64^Cu-dotatate PET computerized tomography (CT) has revolutionized imaging in NENs; however, emerging data suggest that tracer uptake by tumors is not always uniform (Table 4). A retrospective analysis of patients with G1/G2 NETs undergoing peptide receptor radionuclide therapy (PRRT) with ^177^Lu DOTATATE or ^117^Lu-DOTATOC revealed that, compared to tumors with homogenous expression, tumors with heterogenous SSTR expression showed a decreased OS and time to progression (TTP) after PRRT (e.g., TTP 54 months vs. 26 months (*p* < 0.013)) [80]. Kayani et al. reported that ^68^Ga-DOTATATE PET/CT is a superior imaging modality when compared with ^18^F-FDG PET/CT for well-differentiated NETs [81]. They proposed that paired functional imaging with both ^68^Ga-DOTATATE and ^18^F-FDG may provide a more comprehensive tumor assessment in intermediate- and high-grade tumors [81]. Others have reached similar conclusions, that combined imaging with dual tracers (^18^F-FDG PET and ^68^Ga-DOTATATE) may help guide therapy in patients with a G2 or G3 NET at diagnosis, with worse outcomes after PRRT in patients with FDG-avid tumors [11,81,82,83,84]. Zhang et al. performed dual tracer imaging on GEP-NENs of all types (including NETs and NEC) [11]. A negative correlation between ^68^Ga-DOTATATE SUVmax and the Ki-67 index was observed, along with a positive correlation between ^18^F-FDG PET SUV_max_ and the Ki-67 index [11]. While some patients had tumors with concordant update on both types of PET scans, others had tumors that were negative on both scans, positive on only one scan, and/or showed heterogeneous tracer uptake [11]. The overall survival correlated with the imaging findings; tumors that were only ^68^Ga-DOTATATE-negative or only FDG-PET-positive had the worst prognosis.

Chan et al. also performed dual FDG-PET and ^68^Ga-DOTATATE PET imaging on patients with metastatic G1-3 NETs. Patients were classified into five groups (P1–P5) based on the SUV_max_ of the of the lesions on the two types of functional imaging, also known as the “NETPET score”. The NETPET score correlated with survival and may provide information beyond the histological grade [85]. This was again validated in a larger cohort of 319 patients [86]. A similar study demonstrated that a score (FDZ score) based on SUV_max_ assessed by dual imaging correlates with overall survival in G3 GEP-NENs [87]. Another group reported on the use of ^18^F-DOPA PET (18 Fluro-dihydroxyphenylalanine) in functional gastrointestinal NETs. The mean uptake of ^18^F-DOPA varied 1–44-fold (median = 8-fold) between individual lesions within the same patient [88], providing additional evidence of intratumoral heterogeneity at the functional level.

Changes in functional imaging in the setting of drug resistance and grade progression have also been noted. Assi et al. evaluated seven patients with rapid disease progression after PRRT, two of whom had biopsy-proven grade progression (a G1 NET to a G3 NET, and a G2 NET to a NEC). Three patients underwent ^18^F-FDG-PET imaging, all with uptake in some, but not all, lesions. Most patients had a mixed response in ^68^Ga-DOTATATE imaging, with some patients demonstrating variable tumor uptake in either FDG-PET or ^68^Ga-DOTATATE PET imaging [89]. These findings add to the growing body of evidence suggesting intrapatient tumor heterogeneity and changes over time in GEP-NENs.

## 8. Conclusions

De novo NENs consist of both well and poorly differentiated tumors and account for a minority of neoplasms arising in any organ site. Taken together, however, they represent the second most common tumor type in the gastroenteropancreatic system, with G3 NETs emerging as a new and less well-defined subtype, especially in the pancreas. The treatment options for advanced GEP-NETs have greatly improved in the past two decades in parallel with a better understanding of the molecular underpinnings and the discovery of new targeted therapies. However, the timing, choice, and optimal sequence of therapy remain ill defined. Resistance is ubiquitous, and treatment for advanced disease remains palliative.

Intrapatient heterogeneity has been described at the many levels in GEP-NETS, including histopathologic features (including grade and the Ki-67 index), genomic and other molecular changes, the tumor microenvironment, and imaging characteristics [9]. Studies of small-cell lung cancer and treatment-emergent lung and prostate NEC suggest that the neuroendocrine phenotype can evolve over time, even arising in the setting of non-neuroendocrine elements and potentially contributing to resistance [90,91,92]. Whether or not these findings can be extrapolated to GEP-NETs is unknown, but the concept of intratumoral heterogeneity appears to be a common theme across tumor types. Grade heterogeneity exists at baseline in GEP-NETs, particularly within larger primary tumors and in liver metastases compared to primary tumors. Grade can also change over time and with therapy in GEP-NETs, potentially contributing to resistance and presenting barriers to accurate diagnoses, risk stratification, and treatment selection. Current estimates suggest that grade progression is common, occurring in 28–83% of GEP-NETs over time and associated with reduced survival [7,69,75,76]. This finding has important implications for the use of archived tissue for clinical trial eligibility and risk stratification (particularly in panNETs, which appear to have the highest rate of grade progression over time). The mechanisms underlying grade progression and migration are unknown in GEP-NETs, as is whether higher-grade clones exist at baseline or emerge over time. Limited data suggest that progression to G3 (typically well differentiated) from G1/G2 is most common in panNETs [7,75,76].

Clinicians should be aware of the potential for intrapatient tumor heterogeneity, including variable SSTR expression, FDG-avidity, Ki-67 indices, molecular alterations, and grade progression over time (Figure 1). Heterogeneity can have important therapeutic implications, e.g., outcomes with PRRT are worse if the SSTR expression is variable, the tumor has an FDG-avid component, or there is an element of high-grade disease [93]. Platinum-based therapy may be preferable if there is a component of tumor with a Ki 67 index >55% [35,94]. Immunotherapy may play a role if a high tumor mutation burden emerges over time (e.g., after temozolomide-based therapy) [95]. Current data suggest that the early identification, prevention, and/or reversal of grade progression may improve outcomes in GEP-NETs over time. Providers should have a low threshold for repeat tumor biopsies and/or ^18^F-FDG PET imaging in the face of an evolving tumor biology to rule out progression to G3, particularly if the results would impact future therapy or prognoses [75]. Importantly, while the conversion from G1 or G2 NETs to G3 NETs has been repeatedly documented, it is generally believed that NECs do not typically arise from well differentiated NETs [7,73,75,78]. Apparent reductions in tumor grade over time should be interpreted with caution, considering the potential impact of prior treatment, the size of the biopsy, and site of biopsy (i.e., when there is discordance, primary tumors typically demonstrate a lower grade than that of biopsies of metastatic sites). Future strategies may include the incorporation of liquid biopsies, serial tumor biopsies, and/or imaging techniques to identify the most highly proliferative regions for biopsies, recognizing that a Ki-67 index obtained from a routine FNA or core needle biopsy may not adequately represent the whole tumor. In the long term, a greater awareness of tumor heterogeneity and a better understanding of the mechanisms underlying resistance to therapy and the impact of treatment on histopathologic features will help guide a personalized approach to patients with GEP-NETs.

## Figures and Tables

**Figure 1 cancers-15-03712-f001:**
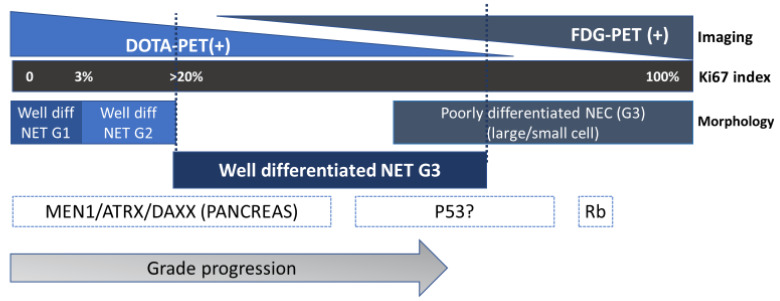
Emerging clinical classification scheme for NENs: integration of pathologic, molecular, and imaging features.

**Table 1 cancers-15-03712-t001:** WHO 2022 classification of neuroendocrine neoplasms of gastrointestinal and pancreatic biliary organs [4].

Terminology	Differentiation	Mitotic Count	Ki-67 Index
Grade 1 NET	Well differentiated	<2/2 mm^2^ and/or	<3%
Grade 2 NET	Well differentiated	2–20/2 mm^2.^ and/or	3–20%
Grade 3 NET	Well differentiated	>20/2 mm^2^ and/or	>20%
Neuroendocrine carcinoma (NEC)	Poorly differentiated Small-cell cytomorphology Large-cell cytomorphology	>20/2 mm^2^ and/or	>20%
Mixed neuroendocrine non-neuroendocrine neoplasm (MiNEN)	Well or poorly differentiated component (≥30%)	Variable	Variable

**Table 3 cancers-15-03712-t003:** Selected studies assessing heterogeneity in metachronous biopsies in GEP-NETs: Ki67 proliferation index and/or grade.

Reference	N	Site of Origin	Definition of Metachronous (mo)	Type of Heterogeneity		Description	Grade Change, (n)
				**Ki67 Index** **N (%)**	**Grade Change** **N (%)**		
Singh [69]	43	NET mix ^^^ (47% SB; 14% P) (baseline grade NOS)	Not explicitly Defined ^&^ (13/16 cases of grade change metachronous; range 229–2613 days between samples) (3/16 cases of grade change synchronous; range 111–143 days)		12/43 (28%)	**Higher grade** over time (9/12 initial biopsies of the primary site; 11/12 follow-up biopsies of the metastatic site)	G1→G2(5) (3 SB, 1P, 1R) G1→G3(5) ** G2→G3(2) **
					4/43 (9%)	**Lower grade** over time (3/4 had biopsies of the metastatic site followed by the primary site; ¼ had serial biopsies of the metastatic site)	G3 → G1 (1) (1 SB) G2→ G1 (3) (1 SB, 1 P, 1R)
Grillo [7]	12	GEP-NET ^#^ (52% SB; 27% P) (93% G1/2)	>4 mo		10/12 (83%)	**Higher grade** over time in metastases	G1→ G2 (9) (NOS) G2→G3 (1) (1 P)
Botling [75]	46	Pancreas NET (96% G1/2)	Not defined *	76% with ↑ Ki67 index	34/59 samples (58%)	↑ Ki67 index +/− **higher grade** in metastases vs. original biopsy (52% with G1/2 to G3 NETs)	G1→G2 (7) G1/G2→G3(27) (all P)
					4/59 samples (7%)	**Lower grade** over time in metastases vs. original biopsies	
Panzuto [76]	43	G1 /G2 GEP-NET ^^ (56% P; 44% SB)	≥3 mo ^&^ (range of 3–148 mo for pancreas; 5–128 mo for SB)	28/43 (65%) with ↑ Ki67 index (71% P; 29% SB)	12/43 (28%)	↑ Ki 67 index and/or **higher grade** at disease progression (10/24 (41.7%) of P cases and 2/19 (10.5%) of the SB cases had grade progression)	G1→G2 (8) (6P; 2 SB) G2→G3 (4) (all P; 4/24, 16.7%)
					2/43 (5%)	**Lower grade** at progression	G2→G1(2) (SB only)
Merola [77]	47	G1/G2 GEP-NET (43% P; 57% SB)	4–176 mo	31/47 (65%) with ↑ Ki-67 index		**Higher grade** in 34% of cases over time	

N = number of patients (unless otherwise specified); mo= months; NET = neuroendocrine tumor; SB= small bowel; P = pancreas; R = rectal; GEP-NET = gastroenteropancreatic neuroendocrine tumors; NOS = not otherwise specified; ^^^ small bowel > pancreas > lung> unknown > large bowel/rectum, appendix, gastric, other; ^#^ pancreas, jejunum/ileum >> stomach, duodenum, colon/rectum; grade at diagnosis-N = 37 G1, N = 19 G2, N = 4 G3; ^^ small bowel, pancreas; ^&^ Initial biopsies mostly from primary tumors; follow-up biopsies mostly from metastases; * 70% of initial biopsies and all follow-up biopsies from metastases; ** 7 cases of progression to G3: 3 SB, 1 P, and 3 bronchial NETs.

**Table 4 cancers-15-03712-t004:** Heterogeneity seen in somatostatin receptors in functional imaging.

Study	N	Tumor Type	Site	Proportion of Patients/Lesions with Heterogeneity	Types of Scans Compared	Conclusions
Chan [85]	62	G1–G3 NET	NET mix (P = 39%, midgut = 32%)	33/62 (53%)	^68^Ga- DOTATATE and ^18^F-FDG PET scan.	Dual imaging is prognostic
Zhang [11]	83	GEP-NEN	P = 27%, GI = 42%, Unknown = 13%	37/83 (44%)	^68^Ga-DOTATATE and ^18^F-FDG PET scan	Dual imaging is prognostic, especially if Ki-67 > 10%.
Kayani [81]	38 (303 lesions)	G1–G3 NEN	Lung = 6 GEP = 28 Unknown = 4	71/303 (23%)	^68^Ga-DOTATATE and ^18^F-FDG PET scan.	Dual imaging facilitates the characterization of intermediate and high-grade NETs (and the heterogeneity within and between tumor sites)
Graf [80]	65	G1/G2 NET	NET mix (ileum = 36.9%, P = 24.6%)	28/65 (43%)	SSTR expression by ^68^Ga-DOTA PET imaging	Heterogeneous SSTR expression by ^68^Ga-DOTATATE or ^68^Ga-DOTATOC PET imaging is prognostic in G1/2 NETs
Zhang [83]	495	G1/G2/G3 NEN	NET mix (P = 199, midgut = 139, rectal = 20, lung = 38, stomach = 8, unknown/other = 91)	382/495(77%)	^68^Ga-DOTATATE and ^18^F-FDG PET scan in patients before PRRT	Positive lesions in ^18^F-FDG PET imaging is an independent prognostic factor in patients treated with PRRT.
Chan [86]	319	G1/G2/G3 Unknown NEN	NET mix (midgut = 52%, P = 36%, hindgut/rectum = 7%, other = 5%)	193/319 (63%)	^68^Ga- DOTATATE and ^18^F-FDG PET scan	Dual imaging as prognostic.
Adnan [84]	36	G1–G3 NET/NEC	NET mix		^68^Ga-DOTATATE and ^18^F-FDG PET scan in patients before PRRT and WHO grading	Dual imaging showed better prognostication in PRRT than WHO grading, differentiation, and immunohistochemistry

NET mix = neuroendocrine tumor mix; P = pancreas; GEP-NEN = gastroenteropancreatic neuroendocrine neoplasm.

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
