# Peer review of "Grade Progression and Intrapatient Tumor Heterogeneity as Potential Contributors to Resistance in Gastroenteropancreatic Neuroendocrine Tumors"

_cancers, 2023, doi:10.3390/cancers15143712_

Round 1

Reviewer 1 Report

Thank you for submission of the article to the journal.

The article is comprehensive and novel, with an interesting compilation of the literature.

My suggestion to further enhance the article would be:

1. to include imaging studies showing FDG and SSR PET/CT.

2. any imaging studies showing heterogeneity predictor

3. please add AJCC /  NANETS / ENETS staging

Thank you

Author Response

Dear Editor,

Thank you for your excellent comments and reviewing the article. See enclosed revised manuscript. In addition, our point-by-point response is below:

Review 1

  1. Include imaging studies showing FDG and SSR PET/CT and any imaging studies showing heterogeneity as a predictor--Heterogeneity related to functional imaging is discussed extensively in the paper; see  lines 409-451 and table 4. The potential implications of heterogeneity on functional dota imaging and/or uptake on FDG PET are discussed.

Please add AJCC /  NANETS / ENETS staging-We have added reference for the AJCC/ENET staging. Given the focus of the paper and the other reviewer’s suggestion to cut back on the background, we hope it will suffice  to reference the different staging systems, rather than describe them in detail .

Reviewer 2 Report

In this well-written review Varghese and colleagues demonstrate that heterogeneity and grade progression are common phenomena in NEN and of prognostic significance.

The first 4 paragraphs contain a lot of general information about NEN but lack the direct connection to the topic you expect by reading the title.

I would encourage the authors to elaborate a bit more to the clinicians what consequences the clinician should draw from being aware of the heterogeneity in NEN (in addition to have a low threshold for re-biopsy). E.g. if heterogeneity in SST PET imaging is associated with shorter PFS after PRRT should another treatment option such as chemo (panNET) or molecular targeted treatments be preferred? Should FDG-PET positivity which is associated with poorer survival and less response to PRRT lead to more aggressive treatment? 

Suggestions regarding wording/ references/typos:

Line 55: rather grade than classification

Line 102: Tyr3-octreotate (177Lu-DOTATATE) (GEP-NETs) : space before first bracket missing and GEPNET should be deleted

Line 128: The prospective trial by Pamela Kunz should be added: (1)

Line 142: The DART trial only reported response in NEC, not in NET patients

Line 144: Poorly differentiated NEC: as NEC are always poorly differentiated (by definition) poorly differentiated can be deleted.

Line 161: Prospective data of second line FOLFIRI by Walter et al. (last ESMO) could be added

Line 166: Although I agree that in the DUNE trial efficacy was limited at least for the NEC subgroup (and only for this subgroup) the primary endpoint was reached.

Line 191/192 rather upgrading and downgrading (not staging)

Line 228: G1/2 NET [3] While   space before bracket/reference missing

Line 234/235: not only outcome is different but also treatment recommendations are

Line 256/257: In a newer publication a grade concordance in 80% of multifocal small bowel NET primaries was reported, reference could be added (2)

Line 273: delete dot after outcome

Reference could be added in the paragraph “heterogeneity between primary and syn met:(3)

Reference could be added in the paragraph grade progression and grade migration over time: (4)

Line 320: an increase (not: and increase)

Line 368: dot after mutations missing

Line 422: Delete dot after uptake (uptake [11].)

Line 460: Dot after reference 9 and in front of “Studies” missing

Line 492: Discovery of newer drugs?

1.     Kunz PL, Graham NT, Catalano PJ, Nimeiri HS, Fisher GA, Longacre TA, et al. Randomized Study of Temozolomide or Temozolomide and Capecitabine in Patients With Advanced Pancreatic Neuroendocrine Tumors (ECOG-ACRIN E2211). J Clin Oncol. 2023;41(7):1359-69.

2.     Jesinghaus M, Poppinga J, Lehman B, Maurer E, Ramaswamy A, Grass A, et al. Frequency and Prognostic Significance of Intertumoural Heterogeneity in Multifocal Jejunoileal Neuroendocrine Tumours. Cancers (Basel). 2022;14(16).

3.     Shi C, Gonzalez RS, Zhao Z, Koyama T, Cornish TC, Hande KR, et al. Liver metastases of small intestine neuroendocrine tumors: Ki-67 heterogeneity and World Health Organization grade discordance with primary tumors. Am J Clin Pathol. 2015;143(3):398-404.

4.     Merola E, Perren A, Rinke A, Zerbi A, McNamara MG, Arsenic R, et al. High rate of Ki-67 increase in entero-pancreatic NET relapses after surgery with curative intent. J Neuroendocrinol. 2022;34(10):e13193.

Author Response

Dear Editor,

Thank you for your excellent comments and reviewing the article. See enclosed revised manuscript. In addition, our point-by-point response is below:

Reviewer 2

  1. The first 4 paragraphs contain a lot of general information about NEN but lack the direct connection to the topic you expect by reading the title- We have cut back on the background.
  2. I would encourage the authors to elaborate a bit more to the clinicians what consequences the clinician should draw from being aware of the heterogeneity in NEN (in addition to have a low threshold for re-biopsy). E.g. if heterogeneity in SST PET imaging is associated with shorter PFS after PRRT should another treatment option such as chemo (panNET) or molecular targeted treatments be preferred? Should FDG-PET positivity which is associated with poorer survival and less response to PRRT lead to more aggressive treatment? Additional information has been added
  3. Line 55: rather grade than classification: CHANGED, in the recent edit the whole line was strike through.
  4. Line 102: Tyr3-octreotate (177Lu-DOTATATE) (GEP-NETs- space before first bracket missing and GEPNET should be deleted-this has been fixed
  5. Line 128: The prospective trial by Pamela Kunz should be added- Reference added (and an old reference removed)
  6. Line 142: The DART trial only reported response in NEC, not in NET patients —per the publication, efficacy has been reported for Cohort 52 of the DART study-G3 NENs, which included well diff NET (patel 2021)-reference added
  7. Line 144: Poorly differentiated NEC: as NEC are always poorly differentiated (by definition) poorly differentiated can be deleted. CHANGED
  8. Line 161: Prospective data of second line FOLFIRI by Walter et al. (last ESMO) could be added – we have added the appropriate references for NET-02 and the Walter study (both irinotecan-based).
  9. Line 166: Although I agree that in the DUNE trial efficacy was limited at least for the NEC subgroup (and only for this subgroup) the primary endpoint was reached- we have added additional information about the DUNE study as suggested.
  10. Line 191/192 rather upgrading and downgrading (not staging) CHANGED
  11. Line 228: G1/2 NET [3] While   space before bracket/reference missing CHANGED
  12. Line 234/235: not only outcome is different but also treatment recommendations are- CHANGED
  13. Line 256/257: In a newer publication a grade concordance in 80% of multifocal small bowel NET primaries was reported, reference could be added (2) ADDED
  14. Line 273: delete dot after outcome- CHANGED, LINE HAS MOVED DOWN
  15. Reference could be added in the paragraph “heterogeneity between primary and syn met:(3)- ADDED in the table and article
  16. Reference could be added in the paragraph grade progression and grade migration over time: (4) CHANGED, ALSO ADDED TO TABLE
  17. Line 320: an increase (not: and increase) CHANGED, LINE 324
  18. Line 368: dot after mutations missing CHANGED
  19. Line 422: Delete dot after uptake (uptake [11].) CHANGED
  20. Line 460: Dot after reference 9 and in front of “Studies” missing CHANGED
  21. Line 492: Discovery of newer drugs? DELETED